# Integrated diversity and network analyses reveal drivers of microbiome dynamics

Rui Guan,[1] Ruben Garrido-Oter[1,2,3]

**ABSTRACT** Microbial communities are key components of ecosystems, where interactions among microbes drive biodiversity and productivity. An increased number of microbiome data sets are available, owing to advances in sequencing; however, standard analyses often focus on community composition, neglecting the complex interactions between co-occurring microbes. To address this, we developed a computational framework integrating compositional and co-occurrence network analyses. We applied this approach to extensive microbial amplicon data sets, focusing on plant microbiota, which typically exhibits high diversity and remains challenging to characterize due to the large number of low-abundance taxa. We show that identifying a subset of representative microbial taxa captures the overall community structure and increases the statistical power. From these taxa, we inferred a large-scale co-occurrence network and clustered microbes with co-varying abundances into units for diversity measurement. This approach not only reduces unexplained variance in diversity assessments but also captures the key microbe-microbe relationships that govern assembly patterns. Furthermore, we introduced a bootstrap- and permutation-based statistical approach to compare microbial networks from diverse conditions. Our method robustly distinguishes meaningful differences and pinpoints specific microbes and features driving those differences. These results highlight the importance of incorporating microbe-microbe interactions in microbiota studies, leading to more accurate and ecologically meaningful insights. Our framework, available as an R package ("mina"), enables researchers to identify condition-specific interactions *via* network comparison and gain a deeper understanding of community ecology. With broad applicability beyond plant systems, this package provides a valuable tool for leveraging microbiome data across disciplines, from agriculture to ecosystem resilience and human health.

**IMPORTANCE** Understanding microbiome dynamics requires capturing not only changes in microbial composition but also interactions between community members. Traditional approaches frequently overlook microbe-microbe interactions, limiting their ecological interpretation. Here, we introduce a novel computational framework that integrates compositional data with network-based analyses, significantly improving the detection of biologically meaningful patterns in community variation. By applying this framework to a large data set from the plant microbiota, we identify representative groups of interacting microbes driving differences across microhabitats and environmental conditions. Our analysis framework, implemented in an R package "mina," provides robust tools allowing researchers to assess statistical differences between microbial networks and detect condition-specific interactions. Broadly applicable to microbiome data sets, our framework is aimed at enabling advances in our understanding of microbial interactions within complex communities.

**KEYWORDS** plant microbiota, diversity analysis, network analysis

Address correspondence to Rui Guan, rg684@cam.ac.uk, or Ruben Garrido-Oter, garridoo@mpipz.mpg.de.

The authors declare no conflict of interest.

See the funding table on p. 14.

10.1128/msystems.00564-25 **1**

Microbes play an essential role in most ecosystems, forming complex communities whose functions influence biodiversity, nutrient cycling, and, through associations with eukaryotic organisms, host health. In plants, roots interact with diverse microbial taxa from the surrounding soil that provide their host with beneficial functions in exchange for organic carbon (1–4). Advances in high-throughput sequencing have made it possible to investigate these interactions at unprecedented scale, generating large microbiome data sets spanning diverse environments, hosts, and conditions (5–12). Traditionally, these data sets are analyzed from a compositional standpoint, focusing on the relative abundances of microbial taxa in each sample. Specifically, alpha and beta diversity metrics identify how communities differ in richness, evenness, or overall composition across varying environmental and host-associated factors. Such approaches have revealed strong compartment- and host-driven patterns in the plant microbiota (10, 13) and have shown evidence of shared taxonomic groups across host species and environments (10, 14–16). However, defining a universal core of shared taxa at fine resolution remains challenging (17, 18), and focusing solely on differences in composition often neglects the underlying interactions among microbes.

Co-occurrence network analysis aims to address this limitation by characterizing how microbial taxa correlate with each other across samples (19–26). In these microbial networks, nodes represent community members, and relationships between microbes are indicated by edges, which are calculated by comparing the covariance of microbes across samples. However, the compositional nature of microbiome data can lead to spurious correlations if not handled properly (27). To mitigate these effects, novel methods with additional data transformation steps have been developed, such as SparCC (28), SPIEC-EASI (29), or integrated workflows, including CoNet (30) or Net-CoMi (31). Despite these advances, network comparisons across conditions often rely on global topological features derived from inferred adjacency matrices, including complexity (32), clustering coefficient (20), density (23, 25, 32), centrality (23, 25, 33), and connectivity (32, 33). However, the calculation of these features is often subject to arbitrary edge-filtering thresholds, such as significance (33, 34), correlation coefficient strength (34), or top-ranking rules (35), which can affect the network topology and pose challenges for the interpretation and cross-referencing of results across studies (36). Furthermore, the statistical tests for comparing these network features are highly dependent on computationally expensive and time-consuming non-parametric permutation procedures (31). Moreover, there is a lack of standardized statistical tools to robustly assess whether observed network differences are significant and to identify which subsets of taxa are most responsible for those differences.

Here, we present a new computational and statistical framework implemented in an R package called "*mina*" (microbial community diversity and network analysis) that integrates compositional analyses with network-based methods, enabling a more nuanced comparison of microbial communities. By applying this approach to the plant root microbiota data set that we integrated from published studies (8–10, 13, 15, 37–39), spanning multiple soil types, host species (including the model *Arabidopsis thaliana*), and distinct root compartments, we show that diversity analysis performed with "*mina*" has an improved performance compared with the available compositional methods. Furthermore, we implemented a permutation method based on spectral distances to statistically assess differences between ecological networks constructed from samples collected under different conditions associated with the same soil (Cologne Agricultural Soil, CAS). Using this approach, we observed novel patterns of network dynamics between compared conditions and identified the distinctive features that contributed significantly to the network deviation.

## RESULTS

### Absence of a core soil, rhizosphere, and root microbiota at low taxonomic levels

To explore the principles governing the assembly of plant root microbiota, we first integrated samples from diverse associated environments into a comprehensive data set. In total, 3,809 bacterial 16S rRNA and 2,232 fungal ITS2 amplicon sequencing samples spanning diverse soil types, host species, and microhabitats were collected (Table S1, Methods). Sequence data were analyzed with standardized quality filtering, followed by error correction using DADA2 (40), leading to the identification of 42,060 bacterial and 9,337 fungal ASVs (amplicon sequence variants, Methods).

Analyses of alpha diversity showed a decrease in the complexity of both bacteria and fungi from the highly diverse soil communities to those found in the rhizosphere, rhizoplane, and root compartments (Fig. S1). This result is in line with previous studies and reflects that the plant recruits a subset of soil community members through a specific process (7–9, 13, 37, 38). Consistent with previous findings, the most abundant bacterial phyla were Proteobacteria, Bacteroidota, Actinobacteriota, and Chloroflexi (Fig. S2a). Analyses of beta diversity revealed a separation between soil and host-associated compartments, as well as marked differences between communities from different host species (Fig. 1a through d). Among the compartments examined, the largest differences were related to an increase in the average relative abundances (RAs) of Proteobacteria from soil (31.54%) to root (60.72%). In contrast, Chloroflexi showed an opposite trend, with average RAs decreasing from 15.80% to 4.61%. For fungal communities, the most abundant phyla were Ascomycota, Basidiomycota, Mortierellomycota, and Olpidiomycota. Members of the Ascomycota dominated the eukaryotic community profiles and showed increased RAs in host-associated compartments (Fig. S2b).

We investigated the prevalence of bacterial and fungal ASVs in our data set and found that the ASV prevalence across samples followed an exponential distribution within each environment, with most amplicon tags observed only in a small subset of samples (Fig. 1c and d). We observed an almost complete absence of core ASVs (i.e., ASVs found in the majority of samples, criteria vary in studies) in our data set. Later, we examined the distribution of taxa at different taxonomic levels in different compartments from each host species, and core taxa were distinguished only at the higher levels, for both kingdoms (Fig. S3). We also compared the prevalence and aRA of microbes associated with a specific type of soil (Cologne Agricultural Soil, Fig. S4) and observed a similar absence of core microbiota at the ASV level. Given the general absence of core bacterial and fungal taxa at the ASV level and the presence of core taxa at higher taxonomic levels when characterized by different subsets of samples, we hypothesized that conditional filtering, including environmental and host-specific factors, partially contributes to the lack of core ASVs. Taken together, these results prompted us to adopt an alternative strategy for feature selection.

Building on a recently published approach that uses abundance–occupancy distributions (41), we ranked all ASVs by their relative abundance and prevalence, and then applied Procrustes Analysis to quantify their contributions to the overall beta diversity (Methods, Fig. S5). From this, we identified 2,047 bacterial and 370 fungal representative ASVs (repASVs). Collectively, these repASVs constituted major portions of the community in each compartment, ranging from 45% to 54% in soil to more than 70% in roots (Fig. 1e and f). A similar trend was also observed across different host species, with minimal variation seen between distinct soil types (Fig. S6). Restricting alpha and beta diversity calculations to repASVs recapitulated the main patterns of community structure captured by all ASVs (Fig. 1a and b; Fig. S1 and S7). In fact, Bray-Curtis dissimilarities were nearly identical between the full set and the repASVs ($M^2 = 0.031$ for bacteria and $M^2 = 0.038$ for fungi), and including only the repASVs reduced the unexplained variance by roughly 8%–9%. Collectively, these results demonstrate that even without a stable "core" at the ASV level, a carefully chosen subset of abundant and prevalent ASVs can represent

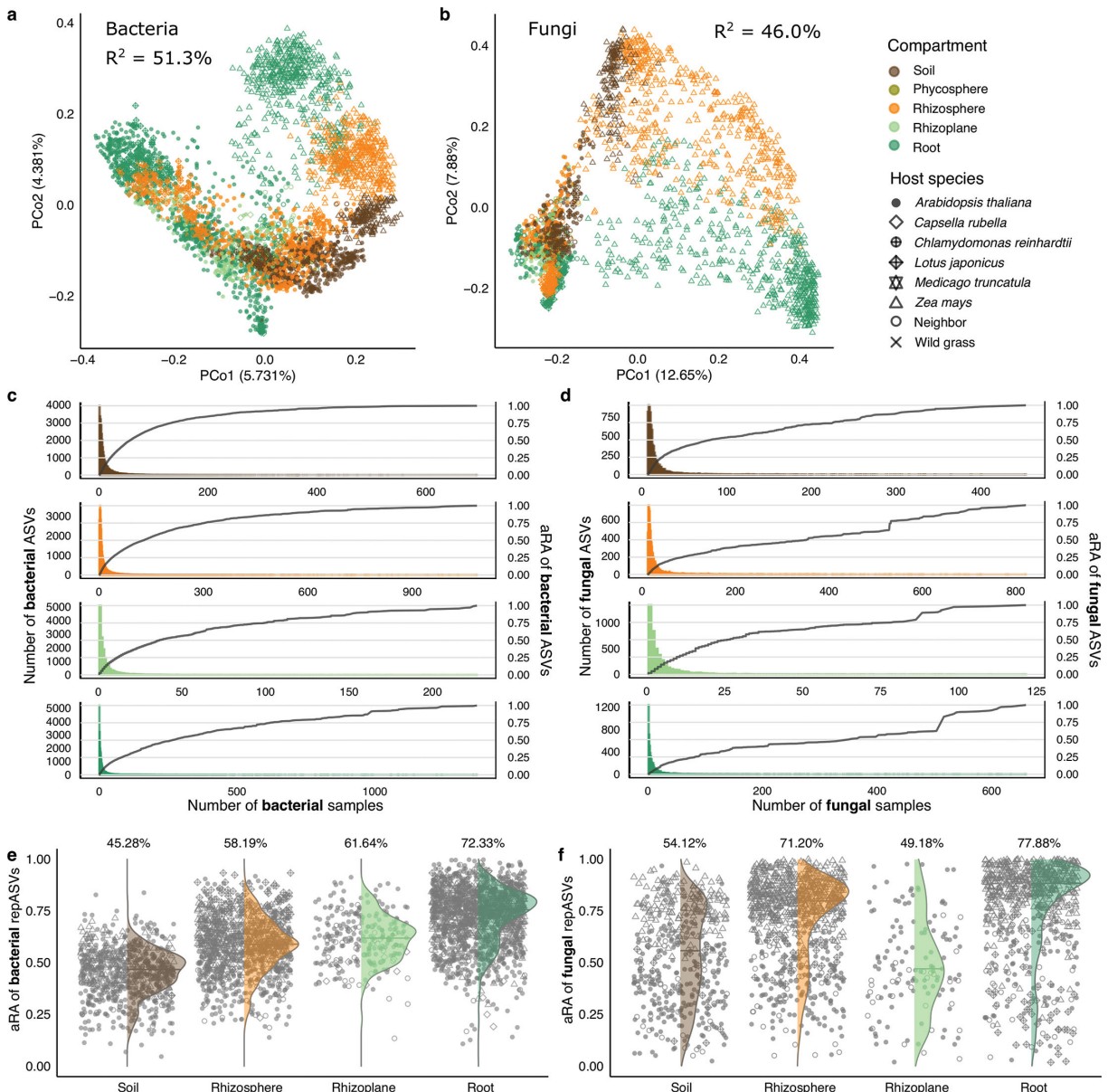

**FIG 1** Plant microbiota diversity and representative members. PCoA of Bray-Curtis dissimilarities between bacterial (a) and fungal (b) communities calculated from all ASVs are shown here. $R^2$ indicates the variance between samples, which cannot be explained by compartment, soil type, host species, host genotype, and experiment condition. Prevalence (bar in different colors with left y-axis) and accumulated relative abundance (aRA, black line in plots with right y-axis) of microbes in root-associated compartments at the ASV level for bacteria (c) and fungi (d) are shown here. Aggregated relative abundance of the selected subset bacterial (e; n = 2,047) and fungal (f; n = 270) ASVs following Procrustes Analysis in each compartment, with multiple host species indicated by the shape of the points. The mean aRA of each compartment is shown.

most of the community's diversity and variation while strengthening the detection of biologically meaningful signals.

## Diversity analyses based on network clusters incorporate microbial interactions

Co-occurrence networks are often employed to explore microbial interaction patterns. Here, we leveraged our representative ASVs, which capture overall community diversity, to develop a network-based diversity index (Fig. 2a; Fig. S8; Methods). First, we generated microbial interaction networks based on pairwise repASV correlation coefficients

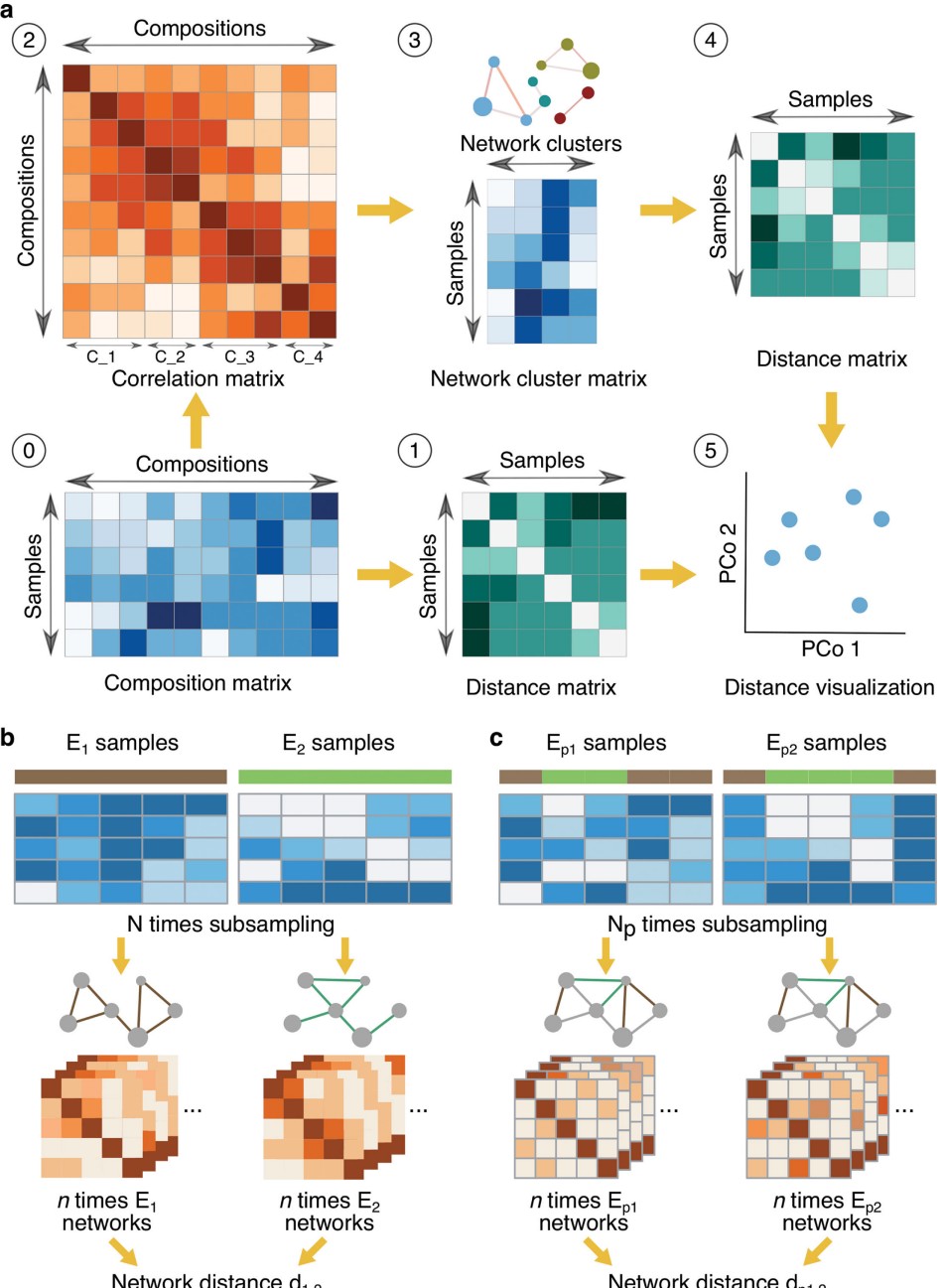

**FIG 2** Novel community diversity and network analysis (*mina*) scheme. (a) Network cluster-based community diversity analysis. Conventionally, a distance matrix (1) is calculated from a composition matrix (0), in which the relative abundance of each community member in each sample is characterized and then visualized by PCoA (5). We introduce here steps (2–4) to first compute the correlation between pairwise community members (2) and then cluster the closely co-varied members, based on which the number or relative abundance is added up as items in the cluster matrix (3). Based on the obtained network cluster matrix, the distance between samples is calculated (4) and visualized (5). C_1, C_2, C_3, and C_4 in (2) are showing an example cluster of samples inferred from step (0). (b, c) Permutation-based network comparison. According to the experimental setup, samples were labeled into different groups in the original ($E_1$ and $E_2$; b) and permutation ($E_{p1}$ and $E_{p2}$; c) data sets. Sample labels in the permutation data set were randomly assigned, that is, pseudo-labels. For each condition, $N$ times subsampling was applied to both the original and permutation data sets. Afterward, correlation networks were inferred as $E_1$ and $E_2$ for the original networks and $E_{p1}$ and $E_{p2}$ for the permutation networks. From this, the network distance ($d_{1,2}$ and $d_{p1,2}$ for the distances between original and permutational networks, respectively) was calculated. The significance of network distance was then evaluated by comparing the distribution of original distances to the distribution of permutational distances between randomized networks using an F-test.

calculated with three methods (Pearson, Spearman, and SparCC). Next, we grouped co-occurring ASVs by clustering nodes in these inferred networks using two algorithms (Affinity Propagation and Markov Clustering; Methods, Tables S2 and S3). After aggregating the relative abundances of each cluster, we calculated Bray-Curtis dissimilarities between samples (Fig. S9). Compared with conventional composition-based analyses, this cluster-based approach reduced unexplained variance by 14%–34% in bacterial communities and 8%–31% in fungal communities. Notably, SparCC networks combined with Affinity Propagation produced the greatest reduction in unexplained variance (20% in bacteria and 15% in fungi; Table S4). At the same time, clear separation of samples according to biological factors (e.g., compartment, host species) persisted, indicating that network-centered diversity indices increased the signal-to-noise ratio in this complex microbiome data set (Fig. S9).

## A statistical framework for network comparison provides insights into microbiota assembly

The main challenge in using co-occurrence networks to compare microbial communities across different environments is determining the statistical significance of observed differences between networks and identifying the underlying drivers of network separation. In microbial ecology, comparison of topological features (e.g., complexity or connectivity) is typically used to assess differences between networks (41). Although alternative approaches that directly quantify structural differences have been developed (e.g., Spectral distance), they are not commonly used for the comparison of microbial networks. A likely reason for this is the lack of methods to assess whether these distances are statistically significant. To address this shortcoming, we have developed a permutation test that can be used to statistically assess the significance of structural differences between microbial networks at the global and local scales, with improved computational efficiency compared with traditional procedure (Methods and Fig. 2b). To perform this test, we first generate observed networks from each condition with bootstrapping and permuted networks by shuffling samples. We then calculate pair-wise distances among the original networks to obtain observed distances and among the permuted networks to obtain a null distribution of distances. Next, we estimate the $P$-value by calculating the proportion of distances between permuted networks that are greater than the observed distances in the original networks, using an F-test. To ensure unbiased results, we compute both observed and permuted network distances from composition count matrices with equal dimensions.

We applied this method to a subset of samples from studies that used the same soil type (Cologne Agricultural Soil; CAS, Table S3) in a greenhouse setting with controlled environmental factors. This subset enabled us to focus on the impact of compartment and host species on microbial network structure variation. Soil-related (soil and rhizosphere) and root-related (including rhizoplane and root) networks show distinct features, especially in the distribution of node closeness and degree (Methods, Table S6; Fig. S10). To statistically evaluate the network differentiation, we employed network comparison analyses using "*mina*." We began by repeatedly subsampling the original data set (33 networks per treatment group, each built from 20 randomly chosen samples) and then calculated the pairwise distances between these subsampled networks. Principal component analyses (PCA) of the distances between subsampled bacterial co-occurrence networks showed a clear separation between environments ($R^2$ = 11.2%; Fig. 3a). This separation, driven by differences in the structure of the microbial co-occurrence networks, was stronger than that obtained from diversity analyses of community compositional data ($R^2$ = 47.4%; Fig. 3b). However, permuted networks generated from randomly selected amplicon samples clustered together ($R^2$ = 80.0%; Fig. 3c), implying that the significance of the observed separation between environments (Fig. 3a) was significant. We also compared the network distances between the networks of different CAS-associated conditions (Fig. S11). Although all the Jaccard distances were significant, due to the highly similar distances between different inter-condition

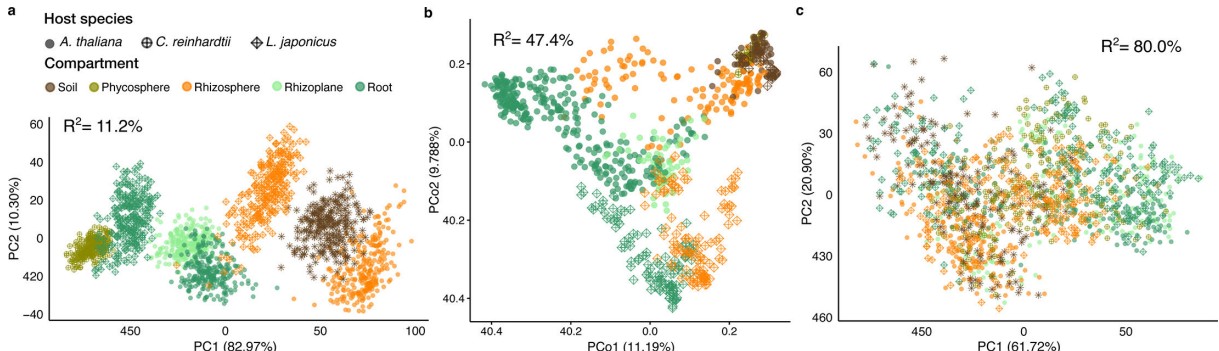

**FIG 3** Network and composition-based comparison between CAS-associated microbiota. (a) PCA of Spectral distances between observed networks inferred from samples of different CAS-related conditions. $R^2$ indicates the variance between networks that cannot be explained by compartment and host species. (b) PCoA of beta diversity between samples. $R^2$ is the ratio of community diversity variance that cannot be explained by compartment, soil batch, host species, and host genotype. (c) PCA of distances between permutation networks inferred from randomly labeled samples of different conditions. $R^2$ indicates the variance between networks that cannot be explained by the compartment and host species. CAS: Cologne Agricultural Soil.

comparisons, this approach failed to characterize the network differences as effectively as the Spectral distance. Applying the same spectral distance analysis to the entire data set yielded similar results for both bacteria and fungi (Fig. S12).

Analysis of network distances (Fig. 3a) showed a separation along the first principal component (82.97% of the variance) between microbial networks derived from soil-based microhabitats (unplanted soil and rhizosphere) and host epi- and endo-phytic

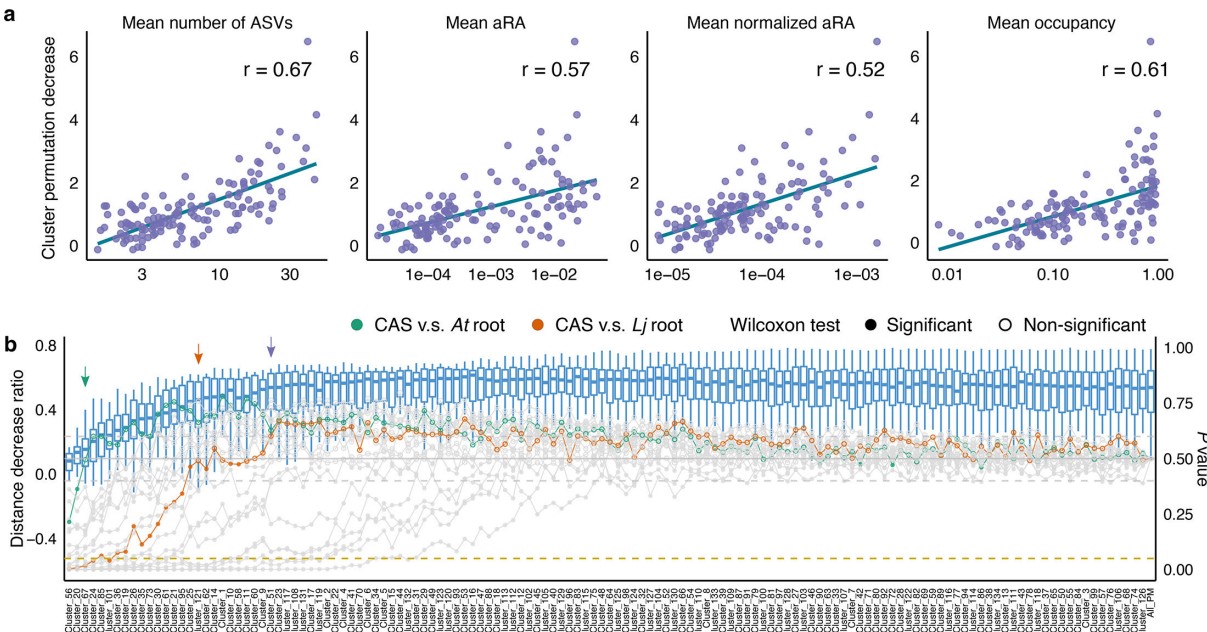

**FIG 4** Clusters of microbes contribute to network separation between different CAS-associated conditions. (a) Correlation between permutation distance change and other features of network clusters. The y-axis shows the average distance change for each network cluster compared with the distance between observed networks inferred from the original data set. The teal line represents the linear regression of the data points. Spearman correlations are computed, and all coefficients are significant ($P < 0.001$). (b) Distance change of cumulative group permutation data sets when nodes are grouped by network clusters. The boxplot shows the distance decrease ratio compared with the average distance calculated by subsampling the original data set (left side y-axis). The line plot presents the *P-value* calculated from the permutation test (right side y-axis). The shape of the points indicates the significance of the Wilcoxon test with FDR correction (full points represent significant comparisons and empty points denote non-significant ones). The dashed line in gold indicates the cutoff for significance ($P = 0.05$). Arrows show the changing point of comparison between specific (CAS vs. *At* root, CAS vs. *Lj* root) or all (in purple) comparisons. *At: A. thaliana, Lj: L. japonicus*; aRA: accumulated relative abundance.

compartments (rhizoplane, root endosphere, and algal phycosphere). This suggests that close associations with a photosynthetic host may lead to significant changes in microbial co-occurrence patterns, potentially driven by higher concentrations of diverse organic carbon compounds or by direct interactions with the host. This separation was more pronounced for *A. thaliana*, where networks derived from root and rhizoplane samples were closer to unplanted soil than to rhizosphere networks (Fig. 3a). Interestingly, this pattern was not observed in the analysis based on ASV relative abundances (Fig. 3b), indicating that the stronger separation between rhizosphere and root compartments can be explained not only by differences in community composition but also by microbe-microbe interactions reflected in differences between microbial co-occurrence networks.

## Permutation analyses reveal microbial modules driving network separation

We next evaluated how specific node groups (at either the family or network-cluster level) influenced the observed differences in network structure. To do this, we adapted our permutation framework so that only the groups under evaluation were randomized (Methods). We then compared the distances among networks generated from partially permuted data with those from the original data set. We assessed two grouping methods: one based on taxonomic level (family) and another derived from network clustering. Overall, most families and network clusters contributed minimally, showing average distance reductions near zero (Fig. S13). However, certain groups had a more pronounced effect, as seen in comparisons between soil and root networks from multiple host species. To understand which characteristics of a group predicted these changes in Spectral distance, we computed correlations with several metrics, including the group size (the number of ASVs contained in the group), aRA, normalized aRA, and occupancy (Fig. 4a; Fig. S14). All metrics were significantly associated with changes in network distance ($P < 0.001$). Of these, group size showed the strongest correlation, followed by occupancy. Interestingly, these relationships were generally stronger for network clusters than for taxonomic families, except in the case of aRA, where results were nearly identical.

To determine how many permutations were needed to eliminate discernible differences between the networks, we systematically randomized node groups cumulatively, after ranking them by their individual impact on network distance (Fig. 4b; Fig. S15). For all tested comparisons and both grouping approaches, we observed an initial surge in distance-change ratios that plateaued after relatively few permutations. The $P$ values (i.e., proportion of permuted distances exceeding the observed ones) hovered around 0.5 once enough groups had been shuffled, indicating that the networks were effectively randomized at this point. To capture when groups began to lose their significance, we introduced the notion of a "changing point," where over half (12 of 21) of the comparisons between conditions became statistically insignificant (Wilcoxon, FDR correction). For families, this threshold corresponded to Chitinophagaceae, which was the 26th group in order of contribution (Fig. S15). The equivalent threshold in network-cluster permutations was Cluster_51, also ranked 26th in their impact on network distance (Fig. 4b). Although both lists contained 26 groups before the changing point, network clusters collectively encompassed fewer ASVs ($n = 736$) than the family-based groups ($n = 985$). Past these changing points, additional permutations did not markedly reduce separation between networks, implying that a small subset of key groups dominates the overall network variance. In line with this observation, plotting PCA of network distances once these critical groups were randomized produced dispersed, random configurations (Fig. S16a and b). Conversely, analyzing only the distinctive groups preserved the main patterns of network differentiation, suggesting that these small sets of discriminative network features carry much of the discriminative power (Fig. S16c and d).

## DISCUSSION

### Representative members of plant microbiota

In this study, we integrated plant microbiota samples from diverse compartments, host species, and soil types for both bacteria and fungi. Our analysis revealed a consistent log-normal distribution of ASVs in each assessed condition, indicating the absence of a well-defined core microbiota. To compensate for the lack of core ASVs, we applied Procrustes Analysis and considered the previously described abundance-occupancy distribution criteria (42, 43) to inform the extraction of representative members. We demonstrate that these representative members recapitulated the majority of aggregated relative abundance (aRA), community diversity, and network dynamics of the whole community system. We propose the term "representative ASVs" (repASVs) and show that their representativeness persists in more focused contexts with reduced technical variation, such as when analyzing only samples from plants grown in the same soil type (CAS-derived samples). Although potential biases may arise from disproportionate representation of certain conditions, our set of repASVs can serve as a baseline for plant microbiota research, particularly for studies with limited sample numbers. Notably, these repASVs are not a fixed set, and we anticipate that they will be refined and updated as new data become available, allowing for continued improvement in our understanding of plant microbiota composition.

### Higher-order feature-based community diversity analysis

The use of operational taxonomic units (OTUs) or amplicon sequence variants (ASVs) as the primary unit for microbial community diversity analysis has been a subject of debate. Although OTUs can reduce the noise introduced by sequencing errors by clustering highly similar sequences (typically at >97% identity) into a unit, OTU assignments for the same taxon can vary across data sets, complicating cross-study comparisons. In contrast, ASVs eliminate this ambiguity by designating each unique sequence as a separate unit, facilitating comparisons between studies. However, the presence of polymorphic copies of marker genes in certain taxa can artificially inflate diversity estimates and introduce misleading correlations in microbial network analyses (44). To address these limitations of traditional approaches, we developed a diversity measurement that groups microbes into clusters based on co-occurrence patterns rather than sequence similarity. This approach provides several advantages, including reduced noise from sequencing errors and the elimination of artifacts introduced by polymorphic copies of amplicon sequences, which are inherently co-occurring. When we applied this network-based clustering to our large-scale plant microbiota data set, unexplained variation in beta diversity decreased, providing a higher signal-to-noise ratio compared with OTU- and ASV-based approaches. Importantly, this improved resolution did not obscure established drivers of community differences; instead, it highlighted underlying associations among microbes that would otherwise not have been identified by traditional compositional methods alone.

### Comparing co-occurrence networks across environments and identifying drivers of network differences

A key challenge in comparing co-occurrence networks across different environments is determining which observed differences are statistically meaningful. To overcome this limitation, we developed a permutation-based framework that employs a spectral network distance metric. This method allows us to measure how network structures diverge and verify the significance of these divergences. The results from PCA and PERMANOVA (Fig. 3) showed that we could link this network-level variability to known experimental factors, validating our approach to evaluate biologically meaningful differences between co-occurrence networks. Furthermore, the addition of a bootstrap step in the permutation test substantially reduced the computational burden typically associated with repeated network inferences.

We applied *"mina"* to a subset of Cologne Agricultural Soil (CAS)-associated samples and found distinct network patterns not captured by standard diversity metrics. Having identified networks that differed significantly, we investigated which community features were driving these differences. By randomizing the metadata for specific groups (e.g., families or network clusters), we revealed how each subset contributed to differences in network architecture. Our findings show that clustering ASVs by correlation-based clusters, similar to microbial guilds (44), offers better resolution than grouping microbes by higher taxonomy in identifying drivers that distinguish networks from different environments. Furthermore, we found a strong correlation between the abundance and prevalence of a group and its impact on overall network divergence. This insight enables the prediction and identification of key players shaping the microbiota assembly and microbial network structure, informs experimental design of synthetic communities, and thereby strengthens the predictive and explanatory power of microbiome research (45, 46).

## Conclusion and future perspectives

In this study, we developed a new approach to microbial community diversity and network analysis, which we made available in an R package called *"mina."* By applying the network cluster-based diversity analysis in *'mina"* to our integrated plant microbiota data set, we demonstrate that this method reduces noise while capturing biologically meaningful variation across environmental samples. Additionally, we introduced a complementary statistical framework for comparing networks, allowing us to identify significant differences between ecological networks and extract relevant community features. Notably, we observed that plant microbiota patterns can differ substantially when analyzed through our network-based approaches rather than solely by traditional compositional methods, providing novel biological insights. The package *"mina"* also allows the visualization of these analyses, enabling more intuitive exploration of microbial interactions, compared with conventional microbial network comparison methods.

Notably, the inference of networks affects both network cluster-based diversity analysis and the identification of features contributing to network differentiation. In addition to the multiple network inference methods implemented in the package, networks pre-constructed using external methods can also be analyzed using *"mina."* Moreover, we recognize the importance of functional properties of the microbiome. By integrating functional prediction methods, such as PICRUSt (47), into our *"mina"* package in future updates, we could provide valuable insights into functional ecology. Although our approach was tested on a large and complex data set from the plant microbiome using only amplicon sequencing data, it is generalizable to any domain where quantitative features must be mapped onto broader interaction networks, opening up applications in various fields of microbial ecology.

## MATERIALS AND METHODS

### Data set integration

We integrated data sets from former studies in the department (8–10, 13, 15, 37–39), all of which were amplified by the same primer set and sequenced with paired-end 300 bp by the same platform, Illumina MiSeq, including bacterial communities with 16S rRNA gene (amplified V5 to V7 regions using primer pairs 799F AACMGGATTAGATACCCKG and 1192R ACGTCATCCCCACCTTCC) and fungal ITS2 (amplified with fITS7 GTGARTCATC-GAATCTTTG and ITS4 TCCTCCGCTTATTGATATGC). To limit the bias between different studies and assist cross-referencing as well as reproducibility, a standardized data processing pipeline was implemented (https://github.com/Guan06/DADA2_pipeline) to process the data the same way.

## Raw data processing and ASV table generation

The bacterial community data processing pipeline is primarily based on DADA2 (v1.12.1, [48]). For each sequencing run, we demultiplexed the samples according to the barcode sequence. For bacterial communities, we truncated forward and reverse reads to 260 and 240 bp, respectively, and filtered with "maxN = 0, maxEE = c (2, 2), truncQ = 2, rm.phix = TRUE." We then estimated error rates of sequences and inferred sequence variants. By merging the forward and reverse reads, ASVs were obtained, and chimeras were removed. Finally, the reads were mapped back to the ASVs, and the corresponding ASV table was generated. By aligning to the Silva database (v138, [49]), we performed taxonomic classification of final ASVs.

We also applied the DADA2 ITS pipeline for fungal communities, starting with mapping primers to the sequencing reads to remove unamplified regions. Afterward, the same processing as the 16S rRNA gene data were implemented for denoising and producing final ASVs. We assigned ASV sequences to taxonomic groups by mapping them to the Unite database (release 04.02.2020, [50]). ASVs, which could not be assigned to any phylum by the Unite database, were extracted and mapped to the nt database (version 2020 Oct. 15, [51]). We identified 83, 50, and 1 ASVs from *Arabidopsis*, *Lotus,* and *Chlamydomonas*, respectively. In addition, we used ITSx (52) to detect the ITS2 region, and 192 ASVs (among which six were host-related ASVs) were detected without ITS2 sequences. During the data processing, we found ASVF_2, which was barely detected in soil, had extremely high relative abundance (49.52% on average) in the root samples from *Lotus* exclusively. Although it was assigned as Basidiomycota, blast results show it mapped to the *Lotus japonicus* chromosome genome; therefore, we removed it for later analysis.

## Community diversity and network analysis with the "*mina*" package

To ensure the repeatability and reproducibility, we integrated the data processing, as well as novel diversity and network analysis methods, into an R package called "mina" (short for microbial community diversity and network analysis; https://bioconductor.org/packages/release/bioc/html/mina.html), which is accepted and maintained by Bioconductor. As indicated by the package name, the "*mina*" workflow could be divided into two main parts: community diversity analysis and network analysis. A data structure object, also named "*mina*," was defined, which contains all the relevant community features as its slots and can be used for all steps in the workflow.

To begin with, "*mina*" expects count data, such as the commonly used OTU or ASV table, to indicate the abundance of each community member in each sample. In addition, a descriptive metadata table, which includes the group information of samples, is required for downstream analysis (e.g., comparison between treatments). Alternatively, the user can also perform individual steps independently on pre-defined feature matrices (e.g., ASV/OTU tables) without using the "*mina*" object. Notably, *the "mina" package is not limited to the application in plant microbiota but is also applicable to all types of microbial community studies.* More details can be found in the manual or vignettes of the package on the Bioconductor website.

## Microbial community alpha and beta diversity analysis

Alpha and beta diversity were performed for both kingdoms' communities. Samples were rarefied to the same sequencing depth to remove the bias introduced by the uneven number of reads. Alternatively, methods such as variance-stabilizing transformation or experimental quantitative approaches can be implemented for normalization (52). Bacterial and fungal samples were rarefied to depths of 1,500 and 2,000, respectively, using *norm_tab*() functions in "*mina*." Afterward, the Shannon index of each sample was calculated using *the diversity*() function from "*vegan*" (53) to characterize the number of community members and the evenness of their relative abundance distribution. We repeated this process 999 times to reduce the random error of rarefaction, and the average was used as the alpha diversities of the corresponding

samples. Bray-Curtis dissimilarity was calculated to evaluate the beta diversity between samples, with the *parDist*() function from the *"parallelDist"* package using 80 threads due to the large matrix size. Unexplained variance ratio was calculated as previously described (54) using *the get_r2*() function in *the "mina"* package.

## Procrustes analysis and representative compositions extraction

Community members with the highest relative abundance and occupancy were extracted as a subset ASV table. Procrustes Analysis was implemented to compare the community shifts when only considering those ASVs. Function *"procrustes()"* from package *"vegan"* (53) was used to calculate the $M^2$ value, the sum of squared distances between paired points in the Bray-Curtis dissimilarity matrices of the compared ASV tables (Fig. S5). ASVs with top *i*% RA and top *j*% occupancy were selected as subset ASVs, and parameters *i, j* were evaluated from 1 to 20 separately. To trade off the community distortion and complexity, parameters that caused the most considerable decrease in $M^2$ when increasing the number of ASVs added to the subset table were chosen (Fig. S5). For the beta diversity analysis based on the representative ASVs, a depth of 1,000 was used for both bacterial and fungal community rarefaction, considering the data loss when extracting repASVs. Mean alpha diversity from 999 times bootstrap was used for visualization. Afterward, rarefied repASV tables were renormalized, and beta diversity was calculated in the same way as before.

## Network inference, clustering, and network cluster-based diversity analysis

The correlation coefficient between repASVs was computed based on their covariance between the samples. The Pearson and Spearman coefficients were then calculated with the *rcorr*() function from the *"Hmisc"* package using the renormalized ASV table after rarefaction. SparCC was inferred using *fastspar* (28, 55) with the raw ASV table without any rarefaction or normalization. For SparCC, 1,000 permutations were implemented to estimate the *P* values of the edges. Non-significant edges (*P* < 0.05) were filtered to obtain the adjacency matrix for later feature computation. Markov Cluster (MCL, [56]) and Affinity Propagation (AP, [57]) were applied for clustering the nodes within each kingdom network using function *net_cls*() in *"mina,"* which was implemented based on the *mcl*() function from the *"MCL"* package and the *apcluster*() function from the *"apcluster"* package. The former method was limited to only positive edges and was applied to networks with the parameter "–I 2.5" for the inflation. For AP, both positive and negative edges were considered during clustering, and "*P* = 0" was applied for the input preference. We also performed a sensitivity analysis to compare the performance of AP and MCL methods at various parameter settings. For AP, we examined the input preference (*P*) and similarity (*q*) over a range of values from 0 to 1, with a step size of 0.1 (Table S2). For MCL, we tested the expansion and inflation parameters from 2 (the default value in the "MCL" package) to 5, with a step size of 0.5 (Table S3). Afterward, Bray-Curtis dissimilarities between bacterial and fungal samples were calculated based on the aRA of network clusters, which were aggregated RAs of repASVs assigned to the same cluster. The unexplained variance between the communities was then calculated with the *get_r2*() function and compared with the ASV-based diversity analysis.

## Network comparison analysis of the whole data set

Samples were assigned to conditions (here, compartments) randomly in the permutation data sets. Networks for each compartment in both original and permuted data sets were inferred with the parameters "g_size = 80, s_size = 50, rm = FALSE, sig = TRUE, bs = 33, pm = 33" using the *bs_pm*() function from *"mina."* The *adj*() function, which infers microbial interaction networks using Pearson, Spearman, and SparCC coefficients, was implemented within *bs_pm*() for efficiency. To validate the stability of results, network inference with more subsampling replicates was performed, with parameters "g_size = 80, s_size = 50, rm = FALSE, sig = TRUE, bs = 99, pm = 99." The distance for pairwise

comparison of compartments was calculated with the *net_dis*() function. We defined two measurements for network comparison at the global level: Spectral and Jaccard distance. For each pairwise comparison, eigenvectors of the networks were calculated from their Laplacian matrix (**L**), which was obtained by subtracting the adjacency matrix (**A**, the correlation network matrix) from the degree matrix (**D**). Based on this, the Euclidean distance between the first *k* (adjust according to the dimension of the correlation matrix) eigenvalues of two Laplacian matrices was computed to obtain the Spectral distance between networks. Alternatively, Jaccard distance is calculated by dividing the sum of the matrix contrast by the sum of the larger absolute values between two correlation matrices. Afterward, the *P*-value was calculated using a permutation test (58), estimating how frequently the distance observed between true networks (*d*) is larger than the distance between the permuted network (*dp*): $P = \frac{N_{d>d_p} + 1}{N + 1}$, where $N_{d>}dpis >$ the times when *d* is larger than *dp* and N is the numbers of all comparisons. Therefore, to obtain a significant distance, a large *N* is needed. By introducing the subsampling step for both original and permutation data sets, the time needed for network inference (*n*) is reduced exponentially. Afterward, PCA of Spectral distance was plotted for visualization.

## Cologne agricultural soil-associated network analyses

Networks for each compartment associated with the host *A. thaliana* were constructed using Spearman's correlation coefficient as the edge weight. We filtered out edges that were not significant (*P* > 0.05 after FDR correction). The unexplained variances of community diversity based on repASVs were calculated with the *get_r2*() function from "*mina.*" When comparing networks from different compartments and host species, the original and permuted networks for each condition were constructed from repASVs. Due to the limitation of sample numbers, parameters "g_size = 40, s_size = 20, rm = FALSE, sig = TRUE, bs = 33, pm = 33, individual = TRUE, out_dir = bs_pm_dir" were used. The network matrices of all conditions inferred from the subsampled samples of both original and permuted data sets were stored in the defined "out_dir." Network distance was then calculated and tested with *net_dis*() from "*mina.*" As mentioned, the subsampling is completely random; therefore, the network distance calculation process is indeterministic. To guarantee reproducibility, a seed is automatically set by default (from *v2.0.0*) unless it is deactivated by the user. However, the results and conclusions from different runs are highly consistent. The partial permutations, including single-group permutation and cumulative-group permutation, were performed by randomly assigning the groups of the samples for specific community members only to select the discriminative features between distinct networks. Afterward, the repASV table would be renormalized, and networks were then constructed for subsequent comparison and statistical tests.

## ACKNOWLEDGMENTS

We would like to thank Prof. Dr. Paul Schulze-Lefert and Prof. Dr. Gunnar W. Klau for their advice throughout the duration of this project. We thank Dr. Yulong Niu for helping with submitting the R package "*mina.*" R.G.O. acknowledges funding from the Max Planck Society and the German Research Foundation under the German Excellence Strategy, EXC number 2048/1 project 390686111.

R.G.O. conceived the project, R.G. conducted the project with the supervision of R.G.O.. R.G. and R.G.O. wrote the manuscript.

## AUTHOR AFFILIATIONS

[1]Department of Plant-Microbe Interactions, Max Planck Institute for Plant Breeding Research, Cologne, Germany
[2]Cluster of Excellence on Plant Sciences, Düsseldorf, Germany
[3]Earlham Institute, Norwich, United Kingdom

## PRESENT ADDRESS

Rui Guan, Medical Research Council Toxicology Unit, University of Cambridge, Cambridge, United Kingdom

## AUTHOR ORCIDs

Rui Guan http://orcid.org/0000-0003-0862-2368
Ruben Garrido-Oter http://orcid.org/0000-0003-1769-892X

## FUNDING

| Funder | Grant(s) | Author(s) |
| --- | --- | --- |
| German Excellence Strategy | 2048/1 | Ruben Garrido-Oter |

## AUTHOR CONTRIBUTIONS

Rui Guan, Conceptualization, Data curation, Formal analysis, Investigation, Methodology, Project administration, Visualization, Writing – original draft, Writing – review and editing | Ruben Garrido-Oter, Conceptualization, Data curation, Funding acquisition, Investigation, Project administration, Resources, Writing – review and editing

## DATA AVAILABILITY

All the data used in this study is published and cited. Scripts used to generate the analyses and figures could be found on GitHub (https://github.com/Guan06/Guan_and_Garrido-Oter_2025). The R package 'mina' can be downloaded and installed from either Bioconductor (10.18129/B9.bioc.mina) or GitHub (https://github.com/Guan06/mina).

## ADDITIONAL FILES

The following material is available online.

### Supplemental Material

**Supplemental figures (mSystems00564-25-s0001.pdf).** Fig. S1 to S16.
**Supplemental tables (mSystems00564-25-s0002.pdf).** Table S1 to S6.

### Open Peer Review

**PEER REVIEW HISTORY (review-history.pdf).** An accounting of the reviewer comments and feedback.

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
