## [Reviewer comments · mSystems]

Integrated diversity and network analyses reveal drivers of microbiome dynamics

Rui Guan and Ruben Garrido-Oter

Corresponding Author(s): Ruben Garrido-Oter, Max-Planck-Institut für Pflanzenzüchtungsforschung

Review Timeline:

Submission Date:	April 24, 2025
Editorial Decision:	May 21, 2025
Revision Received:	July 24, 2025
Accepted:	August 4, 2025

Editor: Jorge Rodrigues

Reviewer(s): The reviewers have opted to remain anonymous.

Transaction Report:

DOI: <https://doi.org/10.1128/msystems.00564-25>

Re: mSystems00564-25 (**Integrated diversity and network analyses reveal drivers of plant microbiota dynamics**)

Dear Dr. Ruben Garrido-Oter:

Both experts in the areas of microbial ecology and bioinformatics have commented on your manuscript and expressed interest in the study, but also suggested clarifications of methods and definitions used throughout the text. Reviewer 1 raised concerns about the lack of use for the core microbiome and the possibility of adopting a false-discovery rate correction when using MINA. Reviewer 2 provided a series of recommendations to improve the flow of your manuscript. I appreciate if the authors could provide supplementary information to guide interested readers to use your R package, like a working flowchart and summary tables of the results provided.

Revision Guidelines

Sincerely,
Jorge Rodrigues
Editor
mSystems

Reviewer #1 (Comments for the Author):

This study presents a novel computational framework that integrates compositional and network-based analyses to investigate plant microbiota dynamics. The work addresses a critical gap in microbiome research by emphasizing microbe-microbe interactions and their role in community assembly. The development of the R package *mina* and its application to a large-scale dataset are commendable. However, I still have some concerns and suggestions as below:

Major concerns:

- 1, The analysis focused on representative ASVs (repASVs) and network clusters to reduce noise and improve interpretability, but the core ASVs can also play very important role in structuring the microbial network. This analysis should omit the role of core species in the microbial interaction.
- 2, While the absence of a core microbiota is highlighted, could environmental filtering or host-specific factors further explain this phenomenon in real?
- 3, How were batch effects (e.g., sequencing runs, experimental protocols) controlled during dataset integration?
- 4, The choice of SparCC, Pearson, and Spearman correlations is justified, but how were thresholds (e.g., $P < 0.05$) selected for edge filtering? Could false-discovery rate (FDR) correction improve reliability?
- 5, How sensitive are network clusters (e.g., Affinity Propagation vs. Markov Clustering) to parameter choices (e.g., inflation value in MCL)?
- 6, While network separation between soil and root compartments is observed, how do these clusters correlate with functional traits (e.g., carbon metabolism, pathogenicity)? Currently, we are more care about the functionality.
- 7, Why was spectral distance chosen over other metrics (e.g., graph edit distance)? A comparison of distance metrics could strengthen the framework.
- 8, How scalable is *mina* for ultra-large datasets (e.g., metagenomes with millions of ASVs)? Are there plans to integrate functional annotation (e.g., PICRUSt2, KEGG) into *mina* to link structure to function?

Minor suggestions:

- 1, Limitations of the study (e.g., reliance on amplicon data, inability to infer causal interactions) should be explicitly addressed.
- 2, A roadmap for refining repASVs as new data emerge would add practical value, and a visual summary of the *mina* workflow (e.g., a schematic) would enhance clarity.
- 3, Fig. 2 legends should explicitly define acronyms like "NCLs" and "aRA" to aid reader comprehension.

Reviewer #2 (Comments for the Author):

The manuscript by Guan & Garrido-Ote presents a well-written and timely contribution that introduces a powerful framework (*mina*) for integrating diversity and network analyses in microbiome studies. I tested the R package and found it intuitive and well-documented. The provided examples are useful, and the methodological pipeline is clearly explained. Here, I recommend a revision focused mainly on improving clarity in certain sections, better defining key concepts, and enhancing the visualization and interpretation of network-related results.

Although the manuscript focuses on plant microbiota, the framework itself is broadly applicable to other systems, as discussed. The current title may unintentionally limit the perceived scope of the study. I suggest adopting a broader title that reflects the general utility of the method, such as: "Integrated diversity and network analyses reveal drivers of microbiome dynamics". This would more accurately communicate the methodological innovation and attract a wider readership.

While the manuscript presents a robust framework for integrating diversity metrics and network analyses, it may not be immediately clear to readers that *mina* does not infer networks directly, but instead analyzes pre-constructed networks (e.g., inferred using SparCC). I recommend clarifying this distinction earlier in the Introduction to help readers understand the scope and required inputs of the package. Additionally, although the manuscript emphasizes network comparisons, no representative network visualizations are provided. Including sample network diagrams and a table summarizing basic network statistics (e.g., number of nodes, edges, average degree, modularity) in the Supplementary Material would improve clarity and support interpretation of structural differences across conditions.

Lines 126-127: The definition of "core taxa" in this section is somewhat vague. Clarifying the criteria used for identifying core members (e.g., prevalence thresholds or occupancy criteria) would strengthen the rationale for adopting the repASV approach.

Lines 351-357: Although the manuscript mentions that datasets were integrated from previous studies, it would be helpful to explicitly list the primers used for both 16S and ITS sequencing (even if shared across studies) and provide corresponding references. This information is important as primer differences can influence ASV profiles and comparability.

Lines 405-417: The Procrustes-based optimization procedure used for repASV selection could benefit from additional clarity, particularly regarding how the optimal (i, j) parameters were chosen. A schematic or worked example in the Supplementary Material would help readers better understand this process.

Lines 421-426: The manuscript indicates that rarefaction was applied before certain steps but not before SparCC inference. As rarefaction remains a debated practice, it would be helpful to briefly justify its use (e.g., to normalize sampling depth for beta

diversity metrics) or mention alternative approaches such as variance-stabilizing transformation or compositional data methods.

Lines 462-464: The authors state that network distance calculations are indeterministic due to random subsampling. Although results appear stable across runs, I recommend including the option to set a random seed (or at least documenting this possibility in the manual) to enhance reproducibility and transparency, especially for users aiming to replicate the analyses.

We would like to thank both reviewers for their careful reading and constructive comments on our manuscript. We have addressed each of these comments thoroughly, performing additional analyses and clarifying several methodological details. We believe these changes have significantly strengthened the manuscript and improved its clarity and reproducibility.

A detailed point-by-point response to all reviewer comments is provided below.

Reviewer #1 (Comments for the Author):

This study presents a novel computational framework that integrates compositional and network-based analyses to investigate plant microbiota dynamics. The work addresses a critical gap in microbiome research by emphasizing microbe-microbe interactions and their role in community assembly. The development of the R package *mina* and its application to a large-scale dataset are commendable.

We thank Reviewer #1 for their positive evaluation and their detailed and constructive comments.

However, I still have some concerns and suggestions as below:

Major concerns:

1, The analysis focused on representative ASVs (repASVs) and network clusters to reduce noise and improve interpretability, but the core ASVs can also play very important role in structuring the microbial network. This analysis should omit the role of core species in the microbial interaction.

We thank the reviewer for this comment. As noted, the majority of analyses were based on representative ASVs (repASVs), which we demonstrated capture key patterns of microbial diversity. In this study, we aim to promote the concept of repASVs over core ASVs, as we found no core set of ASVs consistently present across conditions at the resolution of ASV-level taxonomy. We therefore propose repASVs as a more appropriate conceptual framework for identifying the group of ASVs that can recapitulate the diversity between different conditions. Furthermore, using a defined set of repASVs enables robust network comparisons with statistical testing, as this approach significantly reduces computational demands. Importantly, our methodology does not negate the role of core species in microbial interactions; instead, it implements a rigorous selection methodology (Procrustes analysis) to identify representative taxa and characterize their interactions and network-shaping roles.

2, While the absence of a core microbiota is highlighted, could environmental filtering or host-specific factors further explain this phenomenon in real?

Environmental filtering and host-specific factors likely contribute to the observed absence of a core root microbiota. To validate this, we analyzed the distribution of taxa across samples from subsets of our dataset, focusing on samples from the same host species and compartment (Fig. S3) or environment (e.g., soil type; Fig. S4). The results revealed a consistent absence of core taxa at the ASV level, with a gradual presence at higher taxonomic levels, such as Order or Class. When limiting samples to specific soil types (Fig. S4), we observed the emergence of

core taxa at the Family level. Taken together, we hypothesize that environmental filtering influences this phenomenon, though it is not the sole cause. We now report this analysis on subsetted data and address this point in the revised version of the manuscript (Figs. S3-S4 and lines 120-126). Our findings suggest that the lack of a core microbiota is universally observed and can be partially explained by conditional filtering, such as environmental and host-specific factors, but not entirely.

Figure S3 Density plots for microbial occupancies in different compartments and host species at multiple taxonomic levels. Bacterial (a) and fungal (b) microbes at each taxonomic levels, from ASV to phylum, are shown here. *At*: *Arabidopsis thaliana*; *Zm*: *Zea mays*; *Lj*: *Lotus japonicus*.

Cologne Agricultural Soil

Figure S4 Occupancy and aRA of Cologne Agricultural Soil associated microbiota at the ASV and family levels. Within each condition (host species and compartment), no enrichment of prevalent composition at the ASV level. While at the family level, an increasing number of taxa with high occupancy was found, and they comprised a large portion of aRA, shown by the steep climbing of the black curve in the end. Left y-axis shows the number of taxa in bar plots colored by compartments and right y-axis shows the aRA indicated by the black line. aRA: accumulated relative abundance.

3, How were batch effects (e.g., sequencing runs, experimental protocols) controlled during dataset integration?

Thank you for your comment. To control batch effects during dataset integration, we prioritized experiments processed using the same protocol and sequencing platform and applied a standardized data processing pipeline. However, due to the lack of overlapping samples across experiments, we could not fully correct for batch effects introduced by sequencing runs, which were found to be minimal. To address this limitation, we ensured that when assessing

unexplained variation not attributable to known experimental factors, we accounted for both biological and technical factors (e.g., sequencing run and experimental condition). This approach allowed us to demonstrate differences between conventional methods and the approaches proposed in this study. Furthermore, beta-diversity analysis (Fig. 1a-b) showed minimal batch effects between samples from the same compartment and host species sequenced in different runs. Similarly, alpha-diversity (Fig. S1) and community profiling (Fig. S2) revealed limited variation between samples from the same or similar conditions sequenced in different runs.

4, The choice of SparCC, Pearson, and Spearman correlations is justified, but how were thresholds (e.g., $P < 0.05$) selected for edge filtering? Could false-discovery rate (FDR) correction improve reliability?

We thank the reviewer for raising this important methodological point. Edge filtering is a relevant factor for network-cluster based diversity analysis. In the developed pipeline, we applied $P < 0.05$ as the threshold for filtering out the non-significant edges before running network clustering. To test the reliability of edge filtering threshold, we further included False Discovery Rate (FDR) corrections (Benjamini–Hochberg procedure) to control for multiple comparisons before applying the significance threshold (P -value < 0.05). We found that FDR affect the number of filtered edges to a very limited extend. For example, for bacterial communities, with repASVs as nodes ($N = 2047$) and Spearman correlation coefficient as edges, 3,030,972 edges (density 0.723) will be remained with significance filtering ($P < 0.05$) and 2,945,110 (density 0.703) with adjusted significance filtering ($P < 0.05$ after FDR). A very limited number of edges were filtered after FDR, table below shows more examples of comparisons before and after multiple testing significance adjust.

Table R1 Number of edges and density of bacterial network when applying different edge filtering criteria.

Sig. correction	Sig. threshold	No. of edges	Network density	No. of clusters (AP)
No	0.05	3,030,972	0.723	81
	0.01	2,668,806	0.637	81
	0.001	2,247,360	0.537	81
	0.0001	1,890,996	0.452	81
Yes	0.05	2,945,110	0.703	81
	0.01	2,572,514	0.614	81
	0.001	2,137,080	0.510	81
	0.0001	1,777,744	0.424	81

As shown in the table above, the significance threshold influences the number of remaining edges and alters the network density to a limited extent. Based on this observation, we further investigated how the significance threshold affects network clustering, tested using Affinity Propagation (AP) with the clustering parameter ‘ $p = 0$ ’. All thresholds listed in the table yield consistent clustering results, with the size of clusters (i.e., the number of nodes in each cluster) illustrated in the figure below. In conclusion, the edge filtering threshold does not significantly

impact network clustering and thus does not substantially affect the network-cluster-based diversity analysis.

Fig. R1 Distribution of cluster size of bacterial network. Significant threshold $P < 0.05$ was applied for Spearman correlation coefficient filtering; AP was applied for network clustering.

5, How sensitive are network clusters (e.g., Affinity Propagation vs. Markov Clustering) to parameter choices (e.g., inflation value in MCL)?

We performed a sensitivity analysis comparing Affinity Propagation (AP) and Markov Clustering (MCL) methods at various parameter settings. Results were generally stable across parameter choices in the range of suggested and default parameters for these methods, confirming the robustness of our clustering approach. These results are now summarized in supplementary materials (L459 – 463; Tables S2-3).

We use the inferred bacterial network (node number $n = 2047$) with a significance threshold of $P < 0.05$ as an example. We performed AP clustering method by varying the input preference (p) and similarity (q) parameters, each from 0 to 1 with a step size of 0.1. As expected, the input preference (p) had a substantial impact on the number of clusters, with higher preferences resulting in larger numbers of clusters. Notably, the number of clusters remained relatively stable across varying values of q , indicating that the clustering structure was robust to different similarity thresholds within clusters.

Table S2 Number of clusters in bacterial network inferred by Affinity Propagation using different input preference and similarity. Parameters in bold are default values used in ‘*mina*’.

Parameter p	Range of parameter q	Min. No. of clusters	Max. No. of clusters
NA	NA	81	81
NA	0	81	81
0	0 - 1	81	81
0.1	0 - 1	100	100
0.2	0 - 1	221	221

0.3	0 - 1	609	613
0.4	0 - 1	1222	1231
0.5	0 - 1	1647	1657
0.6	0 - 1	1894	1899
0.7	0 - 1	2001	2004
0.8	0 - 1	2042	2044
0.9	0 - 1	2047	2047
1	0 - 1	2047	2047

In this study, we chose to set the p to NA, allowing the exemplar preference to be initialized according to the distribution of non-infinite values in the correlation matrix. We set q to 0, so the minimum value is used (as shown in the second row of the table above, in bold).

For MCL, we examined effect of varying expansion and inflation parameters on the number and size of clusters. We firstly filtered out negative correlation coefficients, as MCL is designed to work with positive edge weights. We then tested a range of expansion and inflation values from 2 to 5, with a step size of 0.5. As expected, both expansion and inflation affect the number of clusters, to a more subtle extend compared to AP, with higher values resulting in less refined clustering. In this study, we implemented expansion = 2 and inflation = 2.5 as the default parameters in ‘*mina*’ (as shown in bold in the table below).

Table S3 Number of clusters in bacterial network inferred by Markov Clustering with different expansion and inflation parameters. Bold values are default ones used in ‘*mina*’.

Expansion	Inflation	No. of clusters
2	2	4
2	2.5	4
2	3	5
2	3.5	6
2	4	9
2	4.5	9
2	5	9
2.5	2	4
2.5	2.5	4
2.5	3	5
2.5	3.5	6
2.5	4	9
2.5	4.5	9
2.5	5	9
3	2	2
3	2.5	3
3	3	4
3	3.5	4
3	4	4
3	4.5	4
3	5	4
3.5	2	2
3.5	2.5	3
3.5	3	4

3.5	3.5	4
3.5	4	4
3.5	4.5	4
3.5	5	4
4	2	1
4	2.5	2
4	3	3
4	3.5	3
4	4	3
4	4.5	3
4	5	4
4.5	2	1
4.5	2.5	2
4.5	3	3
4.5	3.5	3
4.5	4	3
4.5	4.5	3
4.5	5	4
5	2	1
5	2.5	1
5	3	2
5	3.5	2
5	4	3
5	4.5	3
5	5	3

We have also shown that the number of clusters varies across different correlation coefficients and clustering methods, ranging from 3 to 191 for bacterial networks and 2 to 58 for fungal networks (previous Table S2, now Table S4). Notably, despite the variance in cluster number, all approaches demonstrated a substantial decrease in unexplained variance ratio. Based on these findings, we concluded that adjustments in clustering parameters do not have an extensive impact on the performance of the network cluster-based diversity analysis.

6, While network separation between soil and root compartments is observed, how do these clusters correlate with functional traits (e.g., carbon metabolism, pathogenicity)? Currently, we are more care about the functionality.

We agree with the reviewer that questions about the functionality are important, although it remains a challenge to link community composition with functional traits using only amplicon data. Although direct functional annotation was outside the scope of this study, we recognize its importance and now mention this explicitly in the revised manuscript and discuss of how functional inference methods like PICRUSt2 could be integrated into future versions of our ‘*mina*’ package (lines 342-344 in the section Conclusion and Future Perspectives).

7, Why was spectral distance chosen over other metrics (e.g., graph edit distance)? A comparison of distance metrics could strengthen the framework.

We appreciate this suggestion. Due to the large number of distance calculations required in the significant test, we chose to use the distance indices with lower computational costs, specifically Jaccard and Spectral distances, which can be calculated in $O(n^2)$ and $O(n^3)$ time respectively (where n is the node number). In contrast, other options such as Graph Edit Distance (GED) are NP-hard, often requiring a time complexity of $O(n^4)$ or worse (Bunke 1997).

To illustrate the differences in time complexity of two implemented distances, we conducted a performance comparison by calculating 1,000 comparisons between two $1,000 \times 1,000$ matrices on a MacBook (Apple M1 Pro chip, 16GB memory). The results, measured using the R package ‘*microbenchmark*’, are summarized in Table R2:

Table R2 Computing time required to run 1000 comparisons. Computing time was measured by R package ‘*microbenchmark*’.

Distance	Min.	Lower quartile	Mean	Median	Upper quartile	Max.	No. of evaluation
Jaccard	6.63	8.29	10.76	8.67	10.15	176.55	1000
Spectral	793.73	800.03	849.80	807.81	820.59	2047.59	1000

Although Jaccard distance is computationally more efficient, Spectral distance was primarily used in subsequent analyses due to its superior sensitivity. For instance, when comparing networks from different CAS-associated microbial communities (Fig. 3a), Spectral distances provided a clearer separation between conditions (see Fig. S11) compared to Jaccard distances.

We are open to incorporating additional metrics in future versions of the ‘*mina*’ package, with a clear indication of the computational resources required for such analyses.

8, How scalable is *mina* for ultra-large datasets (e.g., metagenomes with millions of ASVs)? Are there plans to integrate functional annotation (e.g., PICRUSt2, KEGG) into *mina* to link structure to function?

We thank the reviewer for highlighting this point. In the current study, we have successfully applied ‘*mina*’ to microbial datasets ranging from small to large scale (up to 3,809 samples and 42,060 community features). While we have not directly tested ‘*mina*’ on ultra-large datasets containing millions of ASVs, the fact that our software implements parallel versions of the most computationally costly steps, makes it, in principle, scalable substantially larger datasets efficiently. Indeed, we are currently applying ‘*mina*’ to even larger microbiome datasets in ongoing work.

Regarding the integration of functional annotations such as PICRUSt2, we agree this would be valuable, and while such features are not implemented in the current version, we consider these potential extensions in future updates of the package (lines 342-344 in the section Conclusion and Future Perspectives).

Minor suggestions:

1, Limitations of the study (e.g., reliance on amplicon data, inability to infer causal interactions) should be explicitly addressed.

We have now included explicit mention of these limitations in the revised version of the manuscript (lines 339-347). Although we initially developed “*mina*” with amplicon data, the same analytical approach can be applied to different datasets where features are quantified across a gradient of samples. We have now added this to the Conclusion and Future Perspectives.

2, A roadmap for refining repASVs as new data emerge would add practical value, and a visual summary of the *mina* workflow (e.g., a schematic) would enhance clarity.

We thank the reviewer for these suggestions. We are currently exploring approaches to use ‘*mina*’ to update our analyses of repASVs as new (plant or algae) microbiome datasets appear and provide these results to the community, and this is still ongoing work. Furthermore, using pre-defined repASVs on a subset of samples (CAS-related microbiota) shows that the repASVs defined from the integrated dataset remains their representativeness across these subset samples. This increases our confidence in using the repASVs defined in this study without needing to re-refine repASVs for new data, unless the conditions are substantially different from those included in our dataset.

We have also added a visual summary of the ‘*mina*’ workflow to improve clarity (Fig. S8).

3, Fig. 2 legends should explicitly define acronyms like "NCLs" and "aRA" to aid reader comprehension.

We have added a definition of acronyms to all figure captions. Here in Fig. 2, we have modified the abbreviation "NCLs" to "Network clusters" to improve clarity, as we do not frequently use NCLs in our text.

Reviewer #2 (Comments for the Author):

The manuscript by Guan & Garrido-Oter presents a well-written and timely contribution that introduces a powerful framework (*mina*) for integrating diversity and network analyses in microbiome studies. I tested the R package and found it intuitive and well-documented. The provided examples are useful, and the methodological pipeline is clearly explained. Here, I recommend a revision focused mainly on improving clarity in certain sections, better defining key concepts, and enhancing the visualization and interpretation of network-related results.

We thank Reviewer #2 for their positive assessment of our manuscript and the practical recommendations provided.

Although the manuscript focuses on plant microbiota, the framework itself is broadly applicable to other systems, as discussed. The current title may unintentionally limit the perceived scope of the study. I suggest adopting a broader title that reflects the general utility of the method, such as: "Integrated diversity and network analyses reveal drivers of microbiome dynamics". This would more accurately communicate the methodological innovation and attract a wider readership.

We appreciate this insightful suggestion and have broadened the title to better reflect the general utility of our approach. The revised title is now: “Integrated diversity and network analyses reveal drivers of microbiome dynamics.”

While the manuscript presents a robust framework for integrating diversity metrics and network analyses, it may not be immediately clear to readers that *mina* does not infer networks directly, but instead analyzes pre-constructed networks (e.g., inferred using SparCC). I recommend clarifying this distinction earlier in the Introduction to help readers understand the scope and required inputs of the package.

We thank the reviewer for highlighting this potential source of confusion. In the network cluster-based diversity analyses described in the manuscript, we used external packages (e.g., FastSpar) to infer networks, demonstrating that the '*mina*' package can analyze pre-constructed networks generated using external inference methods not currently implemented within the package. Additionally, the '*mina*' package offers methods and functions to infer microbial interaction networks directly using correlation-based approaches (e.g., Pearson, Spearman, and SparCC via the *adj()* function). This is particularly useful for network comparisons between different conditions due to its efficiency in network construction.

We have explicitly clarified this point in the revised manuscript (lines 340 – 342 and 455 - 457) and in the manual of future releases of '*mina*' (from v2.0.0).

Additionally, although the manuscript emphasizes network comparisons, no representative network visualizations are provided. Including sample network diagrams and a table summarizing basic network statistics (e.g., number of nodes, edges, average degree, modularity) in the Supplementary Material would improve clarity and support interpretation of structural differences across conditions.

Following the reviewer's suggestion, we have added representative network feature visualizations and a summary table of network statistics to the supplementary material (Table S6, Fig. S10). We constructed networks for the microbiota associated with the host *Arabidopsis thaliana* that grew in CAS. Specifically, we used samples from compartments including soil, rhizosphere, rhizoplane, and root. We calculated Spearman's correlation as the edge weight and filtered out edges that were non-significant ($P > 0.05$ after FDR correction). We integrated the number of nodes and edges for each condition, as well as network density into Table S6. The distribution of betweenness centrality, closeness centrality, and degree of nodes are integrated in the manuscript as Fig. S10. However, the large number of the nodes and edges in these networks makes it challenging to visualize them, which is one of the motivations behind our development of new visualization tools/functions (e.g., Fig. 3a) for network comparison.

Lines 126-127: The definition of "core taxa" in this section is somewhat vague. Clarifying the criteria used for identifying core members (e.g., prevalence thresholds or occupancy criteria) would strengthen the rationale for adopting the repASV approach.

Thank you for the suggestion. We have now revised this part and explicitly added the criteria in lines 118-119.

Lines 351-357: Although the manuscript mentions that datasets were integrated from previous studies, it would be helpful to explicitly list the primers used for both 16S and ITS sequencing (even if shared across studies) and provide corresponding references. This information is important as primer differences can influence ASV profiles and comparability.

We have added the relevant information regarding the 16S and ITS sequencing primers that were used in the studies that were included in our meta-analyses (see Methods, lines 365-367).

Lines 405-417: The Procrustes-based optimization procedure used for repASV selection could benefit from additional clarity, particularly regarding how the optimal (i, j) parameters were chosen. A schematic or worked example in the Supplementary Material would help readers better understand this process.

We agree with the reviewer that more detailed explanations would be helpful. We applied elbow method here to determine the optimal (i, j) parameters using results of Procrustes Analysis. We now provide the distance change line plot (Fig. S5) to illustrate how we performed the Procrustes method for community member selection and how different (i, j) parameters affect the M^2 value calculated with Procrustes Analysis.

Figure S5 Procrustes distance between the diversity calculated from full and examined subset community members. Subset members of bacterial (A) and fungal (B) communities were chosen and assessed for both RA and occupancy with thresholds from 1% to 20%. Vertical dash lines indicate the determined threshold for representative ASVs. M^2 : Procrustes distance.

Lines 421-426: The manuscript indicates that rarefaction was applied before certain steps but not before SparCC inference. As rarefaction remains a debated practice, it would be helpful to briefly justify its use (e.g., to normalize sampling depth for beta diversity metrics) or mention alternative approaches such as variance-stabilizing transformation or compositional data methods.

We thank the reviewer for pointing out that rarefaction remains debated in microbiome analyses. We have explicitly addressed our rationale for rarefaction in the Methods section and now also mention alternatives such as variance-stabilizing transformations (lines 408-410).

Lines 462-464: The authors state that network distance calculations are indeterministic due to random subsampling. Although results appear stable across runs, I recommend including the option to set a random seed (or at least documenting this possibility in the manual) to enhance reproducibility and transparency, especially for users aiming to replicate the analyses.

We appreciate the reviewer's emphasis on reproducibility. We have explicitly documented that the '*mina*' package provides the possibility to set a random seed (by default), enhancing transparency and reproducibility. This detail is now clearly stated in the revised Methods and the '*mina*' package documentation (lines 484-485).

Re: mSystems00564-25R1 (**Integrated diversity and network analyses reveal drivers of microbiome dynamics**)

Dear Dr. Ruben Garrido-Oter:

Your manuscript has been accepted, and I am forwarding it to the ASM production staff for publication. Your paper will first be checked to make sure all elements meet the technical requirements. ASM staff will contact you if anything needs to be revised before copyediting and production can begin. Otherwise, you will be notified when your proofs are ready to be viewed.

Thank you for submitting your paper to mSystems. I also thank you for allowing me serve as Editor for this important contribution.

Sincerely,
Jorge Rodrigues
Editor
mSystems